# Genome Wide Association Study Identifies Candidate Genes Related to the Earlywood Tracheid Properties in *Picea crassifolia* Kom.

Chengcheng Zhou [1,2,3], Yingtian Guo [1,2,3], Yali Chen [1,2,3], Hongbin Zhang [4], Yousry A. El-Kassaby [5] and Wei Li [1,2,3,*]

1 National Engineering Laboratory of Tree Breeding, College of Biological Sciences and Technology, Beijing Forestry University, Beijing 100083, China; zcc99610@bjfu.edu.cn (C.Z.); gytxiaozi@bjfu.edu.cn (Y.G.); yalichen123321@gmail.com (Y.C.)
2 Key Laboratory of Genetics and Breeding in Forest Trees and Ornamental Plants of Ministry of Education, College of Biological Sciences and Technology, Beijing Forestry University, Beijing 100083, China;
3 The Tree and Ornamental Plant Breeding and Biotechnology Laboratory of National Forestry and Grassland Administration, College of Biological Sciences and Technology, Beijing Forestry University, Beijing 100083, China
4 Gansu Province Academy of Qilian Water Resource Conservation Forests Research Institute, Zhangye 734031, China; hongbinzhang123123@gmail.com
5 Department of Forest and Conservation Sciences, Faculty of Forestry, University of British Columbia, 2424 Main Mall, Vancouver, BC V6T 1Z4, Canada; y.el-kassaby@ubc.ca
* Correspondence: bjfuliwei@bjfu.edu.cn

**Abstract:** *Picea crassifolia* Kom. is one of the timber and ecological conifers in China and its wood tracheid traits directly affect wood formation and adaptability under harsh environment. Molecular studies on *P. crassifolia* remain inadequate because relatively few genes have been associated with these traits. To identify markers and candidate genes that can potentially be used for genetic improvement of wood tracheid traits, we examined 106 clones of *P. crassifolia*, and investigated phenotypic data for 14 wood tracheid traits before specific-locus amplified fragment sequencing (SLAF-seq) was employed to perform a genome wide association study (GWAS). Subsequently, the results were used to screen single nucleotide polymorphism (SNP) loci and candidate genes that exhibited a significant correlation with the studied traits. We developed 4,058,883 SLAF-tags and 12,275,765 SNP loci, and our analyses identified a total of 96 SNP loci that showed significant correlations with three earlywood tracheid traits using a mixed linear model (MLM). Next, candidate genes were screened in the 100 kb zone (50 kb upstream, 50 kb downstream) of each of the SNP loci, whereby 67 candidate genes were obtained in earlywood tracheid traits, including 34 genes of known function and 33 genes of unknown function. We provide the most significant SNP for each trait-locus combination and candidate genes occurring within the GWAS hits. These resources provide a foundation for the development of markers that could be used in wood traits improvement and candidate genes for the development of earlywood tracheid in *P. crassifolia*.

**Keywords:** *Picea crassifolia*; earlywood tracheid; SLAF-seq; SNP; GWAS

## 1. Introduction

*Picea crassifolia* Kom. (Qinghai spruce) is an evergreen conifer native to the Qilian Mountain areas of northwestern China that are known to be among the world's arid and semi-arid mountains. *P. crassifolia* is the most dominant tree species in this mountainous area and acts as a natural green reservoir for regulating and conserving water resources [1–3]. Drought is the primary factor for limiting tree growth in arid and semi-arid regions, greatly influencing xylem anatomical traits [4]. *P. crassifolia* is well adapted to these environmental conditions and produces high quality wood; however, the mechanism of wood formation

under these conditions is still unclear. Studies conducted on *P. crassifolia* sampled from different altitudes of the Tibetan plateau, northwestern China, found that the development of tracheid radial diameter is closely related to temperature and precipitation, and trees could change their internal characteristics to adapt to changing climate [1,5]. Gymnosperms xylem development, especially tracheid radial growth, have a significant influence on wood formation and adaptability under different environments [6]. As tracheid morphology and cell wall structure influence wood and fibers flexibility, interactions among fibers, as well as the mechanical, physical, and optical properties of the end-products, are important [7]. In *Pinus tabuliformis* Carrière and *P. crassifolia*, the variation in tracheid diameter affects the duration of cell enlargement, and in turn influences their ability to adapt to drought [4]. However, no attempts have been made to explore the genetic factors underlying wood tracheid development variation. Thus, wood traits improvement and adaptability enhancement under harsh environments have become selection targets of tree breeding programs [8]. Therefore, uncovering the genetic basis of *P. crassifolia* wood tracheid traits is relevant for exploring the molecular mechanism of wood development in response to environmental changes.

The morphological characteristics of wood tracheid in conifers species, such as length, width, length–width ratio, wall thickness, lumen diameter, and wall–lumen diameter ratio, are an important basis for wood fiber utilization and key indicators in wood formation and development [9]. These are complex quantitative traits that are substantially affected by environmental factors, even though they are mainly controlled by genetic factors [10]. Although an earlier study identified quantitative trait loci (QTLs) for wood density variation in loblolly pine using restriction fragment length polymorphism (RFLP) marker genotypes [11], but previous methods were partially translated or not implemented in practical tree breeding due to QTLs are often family specific, generally explaining a small amount of phenotypic variation, and the need for very large QTL mapping families to recover desirable combinations of QTL alleles for more than five or six loci [12–14]. Fortunately, it has achieved success in some forest breeding that using genome wide association analysis (GWAS) to generally detect (as could be expected) markers closely related to the target traits with small effects at population level (across many families) in *Populus trichocarpa* Torr. & A.Gray ex Hook [15], *Cryptomeria japonica* (Thunb. ex L.f.) D.Don [16], *Eucalyptus* [17], white spruce [18], and oil palm [19], suggesting that GWAS is an appropriate research method to understand the genetics of complex traits in woody species [20,21].

Previous GWAS have mostly been based on SNP array technology, which can detect known SNP loci, however, new loci could not be accommodated [22]. In light of this apparent limitation, a high-throughput sequencing-based technology known as specific locus-amplified fragment sequencing (SLAF-seq) to offer ample SNP loci for such GWAS analyses is expected to overcome this limitation [23]. GWAS based on SLAF-seq has a series of advantages, including generating high-density SNP loci numbering in millions and detecting novel SNP loci in unknown mutation-harboring loci compared with SNP arrays [24]. Therefore, SNP loci generated from GWAS based on SLAF-seq provides an effective way to understand the genetic mechanism of *P. crassifolia* wood tracheid traits. More specifically, we developed SNP markers of these clones at the whole genome level using SLAF-seq to genotype *P. crassifolia* population comprising of 106 clones and utilized GWAS to determine those loci underpinning wood tracheid traits. The objectives of this study are to: (1) determine the genetic structure of this population, (2) identify SNP loci associated with wood tracheid traits, and (3) explore the candidate genes associated with wood tracheid traits. These results are expected to provide useful information for *P. crassifolia* breeding improvement.

## 2. Materials and Methods

### 2.1. Plant Materials

A diverse collection of 106 *P. crassifolia* clones was used for this study (Table S1). All 106 clones were 20 years old (planted in 1999 and sampled in 2019) growing in the Longqu

National Improved Variety Base, Zhangye City, Gansu Province, China (100°13′42″ N, 38°48′41″ W). These clones were members of seven different clonal seed orchards and were classified according to their seed orchard sources: XS: Xi Shui, LC: Lian Cheng, DHS: Da Huangshan, HX: Ha Xi, DDS: Dong Dashan, LCH: Long Changhe, GC: Gu Cheng, DHK: Da Hekou, QL: Qi Lian, XYH: Xi Yinghe, and SDL: Shi Dalong. Within each orchard, clones were planted at $5 \times 5$ m within and between rows on the same soil type and managed similarly in a complete randomized block design with 18 replications.

## 2.2. Wood Tracheid Traits Phenotyping

A set of 14 wood tracheid traits were phenotyped and used as the research targeted traits (Table 1). From every clone, bark to pith 5 mm wood cores were extracted from the south and north directions at 1.3 m height. Tracheid length, diameter, and lumen diameter were measured by Motic 2.0 software under light microscope. A total of 15–20 tracheids were measured for each clone. Additionally, tracheid wall thickness, tracheid lumen–diameter ratio, tracheid wall–lumen ratio, and tracheid length–diameter ratio were calculated using software EXCEL 2020.

**Table 1.** List of the phenotypes, their abbreviations, and measurement unit.

| Phenotype | Abbreviation | Unit |
|---|---|---|
| Earlywood tracheid length | EL | μm |
| Earlywood tracheid diameter | ED | μm |
| Earlywood tracheid lumen diameter | ELD | μm |
| Earlywood tracheid wall thickness | EWT | μm |
| Latewood tracheid length | LL | μm |
| Latewood tracheid diameter | LD | μm |
| Latewood tracheid lumen diameter | LLD | μm |
| Latewood tracheid wall thickness | LWT | μm |
| Earlywood tracheid lumen–diameter ratio | ELDR | |
| Earlywood tracheid wall–lumen ratio | EWLR | |
| Earlywood tracheid length–width ratio | ELWR | |
| Latewood tracheid lumen–diameter ratio | LLDR | |
| Latewood tracheid wall–lumen ratio | LWLR | |
| Latewood tracheid length–width ratio | LLWR | |

Phenotypic traits indexes (mean value, standard deviation, coefficient of variation, skewness, kurtosis, and other parameters) were analyzed by SPSS 19.0 statistical software. The Origin software (version 8.2; https://www.originlab.com/https://www.originlab.com/, accessed on 20 August 2021) was used to display the frequency distribution of each trait. Correlations among traits were analyzed and visualized by psych and ggplot2 packages in R software 4.1.0.

## 2.3. DNA Isolation, Construction of SLAF Library, and Genome Sequencing

Total DNA was extracted from young needle tissues of each clone using a modified CTAB method [25]. Extracted genomic DNA concentration exceeded 20 ng/L, meeting the required quality for database construction. Due to the lack of *P. crassifolia* (L.) H.Karst. genome sequence, *Picea abies* genome was used as a reference genome (ftp://plantgenie.org/Data/ConGenIE/Picea_abies/v1.0/, accessed on 12 December 2020). The reference genome was 12 G in size with 37.88% GC content. The digestion of genomic DNA and the construction of SLAF-seq library according to the protocol by Biomarker Technologies Co. (Beijing, China). Clusters of libraries were loaded into an Illumina HiSeq for paired-end sequencing. The obtained clean reads were compared to the reference genome by BWA software [26], and the number of SLAF-tags and polymorphic SLAF-tags were counted. SNP markers were developed by GATK [27] and Samtools [28]. Afterwards, SNP heterozygosity was calculated

by PLINK software [29], the final SNP data with MAF < 0.05, heterozygosity > 0.02, and SNPs with more than two alleles were used to filter out.

### 2.4. Population Genetic Analyses and Linkage Disequilibrium

Based on the high-quality SNP markers, the software MEGA X [30] was used to construct the NJ tree among the 106 *P. crassifolia* studied clones. Principal component analysis (PCA) and calculation of a relative kinship matrix were performed using the software EIGENSTRAT [31] and GCTA [32], respectively. Additionally, population structure analysis using 12,275,765 SNPs to infer the genetic background of clonal cluster membership under a given number of populations (K) was carried out. The number of genetic clusters was predefined as K = 1–10 for all clones and was calculated using Admixture software [33,34].

Linkage Disequilibrium (LD) was estimated by calculating the squared allele frequency correlation coefficient ($r^2$) between pairs of all SNP markers distributed throughout the genome using the vcftools software package [35]. The $r^2$ values were plotted against corresponding genetic distances in pairwise distance (bp). A nonlinear fitted-curve was drawn using second-degree locally weighted polynomial regression (LOESS) by applying the "loess" function in the R statistical program (http://www.r-project.org, accessed on 20 August 2021).

### 2.5. Association Genetics Analysis

A total of 12,275,765 SNPs from 106 clones were used in the GWAS. The efficient model was performed with both general linear model (GLM) and mixed linear model (MLM) using TASSEL [36], FaSTLMM [37], and EMMAX [38]. The population structure matrix generated from Admixture was used as the Q matrix for the GLM model, while the Q values and the K values of the kinship coefficient matrix were calculated by SPAGeDi [33] software using MLM model. *p*-values of $p \leq 1.268 \times 10^{-7}$ ($p = 0.01/n$; $n$ = total markers used, which is roughly a Bonferroni correction, corresponding to $-\log_{10}(p) = 7$) and $p \leq 1.268 \times 10^{-8}$ ($p = 0.1/n$; $n$ = total markers used, which is roughly a Bonferroni correction, corresponding to $-\log_{10}(p) = 8$) were defined as the genome wide control threshold and suggestive threshold, respectively. A standard interval of 100 kb (50 kb upstream and downstream) was explored for each candidate locus and adjusted according to the extent of local linkage disequilibrium with the candidate SNP ($R^2 < 0.8$). All candidate genes were annotated by GO, KEGG, NR, KOG, Pfam, and Swissprot databases. Manhattan plots were generated using the R package "CMplot" (https://github.com/YinLiLin/R-CMplot accessed on 20 August 2021).

## 3. Results

### 3.1. Sequencing Results

The SLAF-seq sequencing, resulted in a total of 1375.57 Mb of clean data from the 106 *P. crassifolia* clones. GC content ranged from 39.19 to 43.62%, with an average of 40.27%. The sequencing quality value Q30 ranged from 90.97 to 97.82%, with an average of 95.51% (Table S2). SNP detection was performed, based on 4,058,883 SLAF tags detected in the sequencing, producing a total of 12,275,765 SNP loci with a minor allele frequency >0.05 were generated, and SNP integrity of each sample ranged from 15.40 to 36.56%, and the SNP heterozygosity ranged from 5.41 to 10.99% (Table S3). Our analysis yielded a total of 1,573,899 polymorphic SLAF markers with a mean depth of 21.21 (Table S4). After quality control, a total of 12,275,765 SNPs were used for subsequent GWAS analyses.

### 3.2. Phenotypic Characterization of 14 Wood Tracheid Traits

Tracheid phenotypic data exhibited abundant variation among clones as the 106 clones originated from different forest regions (Figure 1). Coefficients of variation (CVs) of the 14 traits exhibited values ranged between 5.62 and 89.47%, with the largest and smallest CV values were observed for ELDR, and EWLR, respectively, and average CV values across the 14 traits was 17.54% (Table 2). The average CVs of earlywood tracheid properties were

relatively large, indicating that earlywood tracheid properties are easily affected by the environment. In general, the coefficient of variation in a half of 14 wood tracheid traits were more than 10%, indicating the present of large phenotypic variation among the studied clones.

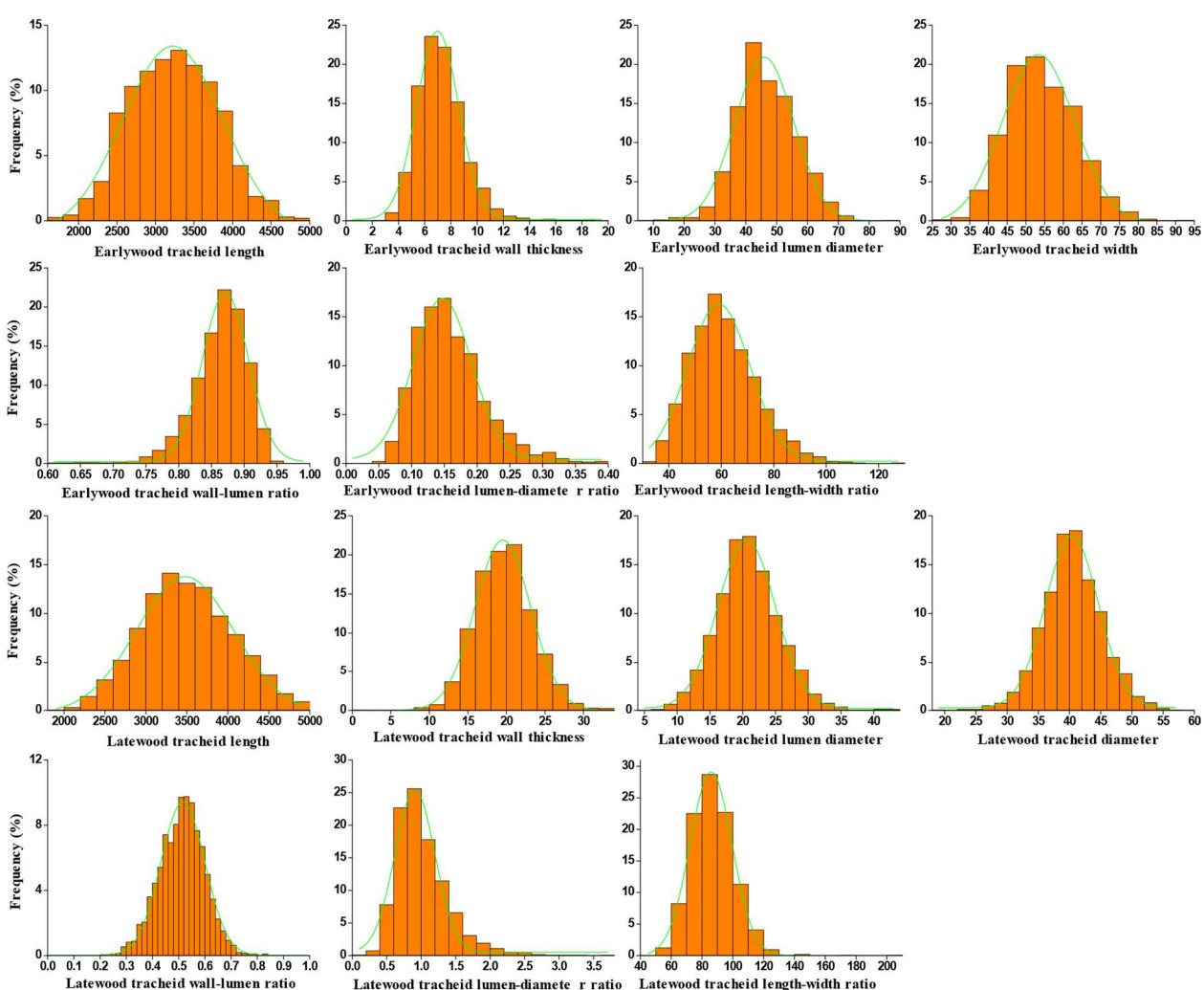

**Figure 1.** Frequency distribution of 14 wood tracheid traits in 106 clones of *P. crassifolia*. The green lines indicate fitting curve using GaussAmp Non-linear fitting model.

**Table 2.** Phenotype statistics of traits in *P. crassifolia* clones.

| Trait | Min | Max | Mean | SD | CV (%) | Skewness | Kurtosis |
|---|---|---|---|---|---|---|---|
| EL | 2303.61 | 4131.40 | 3237.20 | 348.74 | 10.77 | −0.42 | −0.01 |
| ED | 44.37 | 66.46 | 54.14 | 5.26 | 9.71 | −0.40 | 0.36 |
| ELD | 20.17 | 59.62 | 46.78 | 5.97 | 12.76 | 2.71 | −0.42 |
| EWT | 5.28 | 30.29 | 7.36 | 2.40 | 32.64 | 80.35 | 8.41 |
| ELDR | 43.44 | 76.76 | 60.98 | 6.99 | 11.47 | −0.38 | −0.03 |
| EWLR | 0.41 | 0.91 | 0.86 | 0.05 | 5.62 | 69.75 | −7.56 |
| EDLR | 0.10 | 1.74 | 0.17 | 0.16 | 89.47 | 98.73 | 9.77 |
| LL | 2434.23 | 4317.94 | 3505.45 | 339.43 | 9.68 | 0.28 | −0.21 |
| LD | 34.20 | 47.83 | 40.45 | 2.57 | 6.34 | 0.39 | 0.07 |
| LLD | 14.82 | 26.64 | 20.83 | 2.45 | 11.76 | −0.33 | −0.01 |
| LWT | 15.21 | 23.69 | 19.62 | 1.90 | 9.66 | −0.29 | −0.26 |
| LLDR | 67.79 | 101.16 | 87.21 | 6.59 | 7.56 | 0.13 | −0.44 |
| LWLR | 0.40 | 0.62 | 0.51 | 0.04 | 8.71 | 0.10 | −0.04 |
| LDLR | 0.63 | 1.67 | 1.02 | 0.20 | 19.47 | 0.75 | 0.72 |

The absolute values of skewness and kurtosis of most tracheid traits in the studied population were less than 1, and most traits were normally distributed, indicating that these traits were quantitative traits controlled by multiple genes. Correlation analysis showed a highly significant relationship existed mainly among earlywood tracheid traits (Figure 2). In earlywood, significant correlations were observed among ED, EL, ELD, EWLR, ELWR, and EWT, ranging from −0.57 to 0.94 ($p < 0.01$), whereas ED was significantly positively correlated with ELD (traits correlation coefficient ($r_t$) = 0.94, $p < 0.01$), indicating that the diameter and lumen diameter of tracheid in development were increased simultaneously. Additionally, EWT was significantly negatively correlated with EWLR ($r_t$ = −0.86, $p < 0.01$), suggesting that tracheid wall thickness decreased as the lumen diameter of tracheid in earlywood increased. Finally, similar correlation results were observed among latewood tracheid traits.

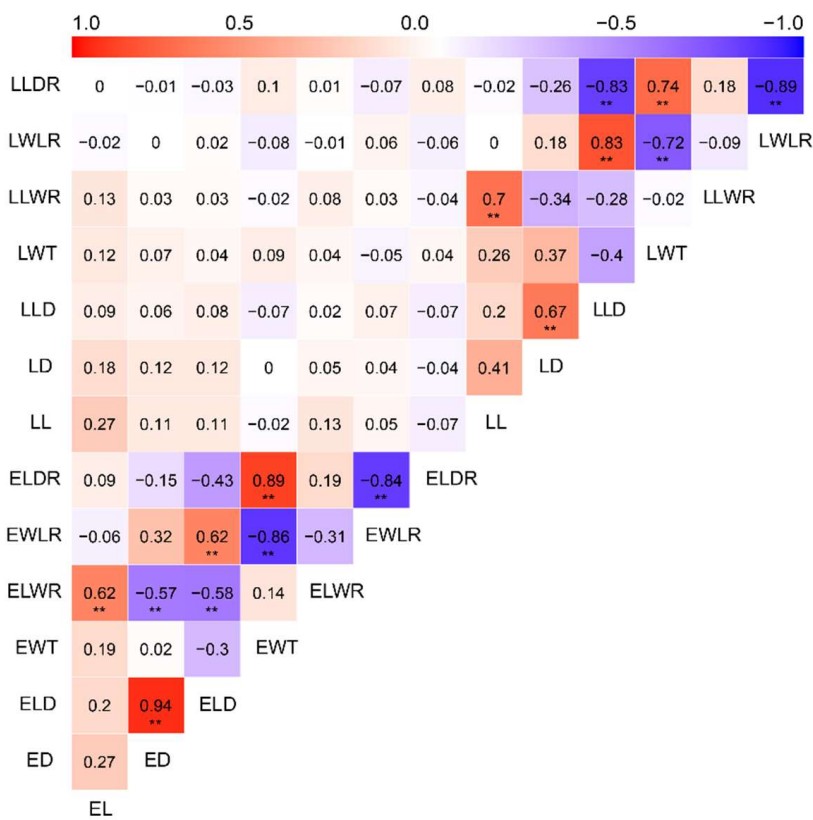

**Figure 2.** Correlation analysis of 14 wood tracheid traits. ** indicates that the correlation was significant at the $p < 0.01$ level. E and D indicate earlywood and latewood, respectively. L indicates tracheid length; D indicates tracheid diameter; LD indicates tracheid lumen diameter; WT indicates tracheid wall thickness; LWR indicates tracheid length–width ratio; WLR indicates tracheid wall–lumen ratio; and LDR indicates tracheid length–diameter ratio.

### 3.3. Analysis of Population Structure and Linkage Disequilibrium

The Admixture software was used to analyze population clustering and structure of the 106 clones (Figure 3a,b). Specifically, clustering was first performed assuming that the number of clusters (K) was between 1 and 10. Then, the results were cross-validated to determine that the optimal K-value was 1 (according to the valley of the error rates of cross-validation). In other words, our results implied that the collection most likely originated from the same ancestors. Principal component analysis showed that the first two principal components explained only 2.86% and 2.63% of the total variance (Figure 3c). The top 18 PCA components cumulatively contribute about 30% of the total marker variation. This means that the population structure of the association panel was weak and the population structure cannot be explained by a few principal components, which might be attributed to

the extensive exchanges of *P. crassifolia* breeding materials. Additionally, we observed family relationships along the diagonal with a scattered distribution of closely related individuals, and the remained part of the relationship matrix indicated low kinship (Figure 3d).

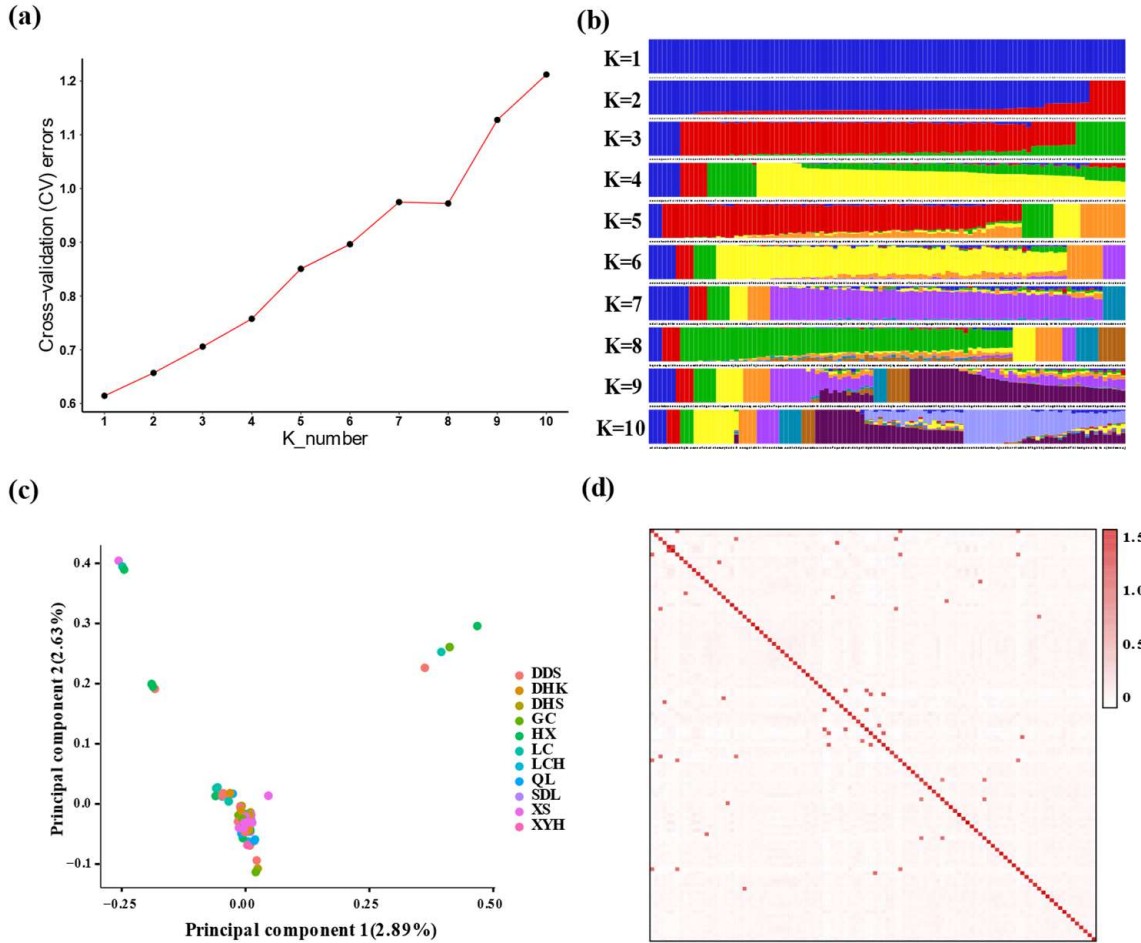

**Figure 3.** The analysis of population structure. (**a**) Population structure plot (ordinate represents cross-validation error rate (CV-value); *x*-axis represents number of clusters (K)). (**b**) Population clustering analyses by admixture software. (**c**) Principal component analyses. (**d**) Heatmap showing pairwise Kinship matrix. The *x*-axis and *y*-axis in QQ-plots indicate the kinship value.

The LD in *P. crassifolia* was estimated using all squared correlation coefficient ($r^2$) values and the physical distances between the same SNP pair. The nonlinear fitted-curve indicated that the LD is low in *P. crassifolia*, rapidly decaying by over 50% (from 0.50 to 0.10) (Figure 4). The average distance associated with the LD decline for $r^2$ = 0.02 varied roughly from 50 to 3000 bp. This result is expected and consistent with the trend of rapid LD decay in conifers, including Norway spruce [14] and loblolly pine [39]. Additionally, we detected high LD extending up to 100 kb. In woody plants, especially conifers, there are relatively high LD exist, for example, LD extend up to 140 kb and >145 kb in Norway spruce [14] and *Shorea platyclados* Slooten ex x Endert [40], respectively. This indicates that associations between phenotypic traits and markers in LD can be more easily and feasibly detected with GWAS than with analysis of quantitative trait loci (QTLs) [40].

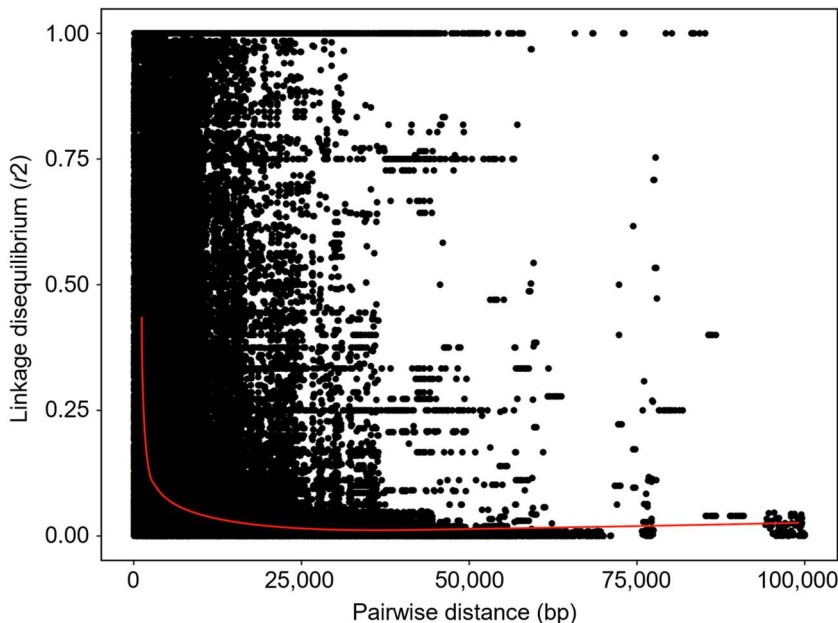

**Figure 4.** Linkage disequilibrium (LD) decay in the 106 *P. crassifolia* clones. Pairwise LD ($r^2$) values plotted against the physical distance (bp) between all pairs of SNPs. The trend line of the nonlinear regressions against physical distance is given by the red line.

*3.4. Association Analysis of 14 Wood Tracheid Traits*

The GLM and MLM models identified more significant SNPs for the former than the later with some SNPs overlapped between the two models. However, the accuracy of MLM was better than that of GLM. A total of 96 SNP loci ($p < 1.11 \times 10^{-8}$) randomly distributed on 887,836 loci (Tables S5 and S6) of *Picea abies* genome [41] were identified on the MLM as significantly associated with three wood tracheid traits (ELDR, EWLR, and EWT). All SNP loci contributed to more than 10% to phenotypic variation. Among them, 9 were simultaneously detected for ELDR, EWLR, and EWT, and 11 SNP were simultaneously detected for EWLR and EWT. These significant associations could reflect that the genetic basis of the observed correlations among these traits, supporting the observed phenotypic correlation and pleiotropic effect between phenotypes.

The regional Manhattan plots and quantile–quantile plots (QQ plots) of GLM and MLM for three earlywood tracheid traits (ELDR, EWLR, and EWT) and one latewood tracheid trait (LWLR) are presented in Figure 5. These plots include candidate genes within 100 kb of the significant SNP marker (50 kb upstream, 50 kb downstream), yielding a total of 67 candidate genes (Table 3), including 33 with unknown functions and 34 with predicted function annotation. Among these SNPs, some were highly associated with LWLR for the GLM, but not with the MLM. Three earlywood tracheid traits (ELDR, EWLR, and EWT) were associated with seven candidate genes (MA_10430313g0010, MA_10429843g0020, MA_10434936g0010, MA_11137g0010, MA_119933g0010, MA_17692g0010, and MA_19953g0020). The synonymous SNP MA_119933 is located on the gene MA_119933g0010 homologous with wall-associated receptor kinase, this ability to bind and respond to several types of pectins correlates with a demonstrated role for WAKs in both the pathogen response and cell expansion during plant development [42]. Therefore, the gene homologous with MA_119933g0010 in *P. crassifolia* may be involved in the cell wall development in earlywood tracheid.

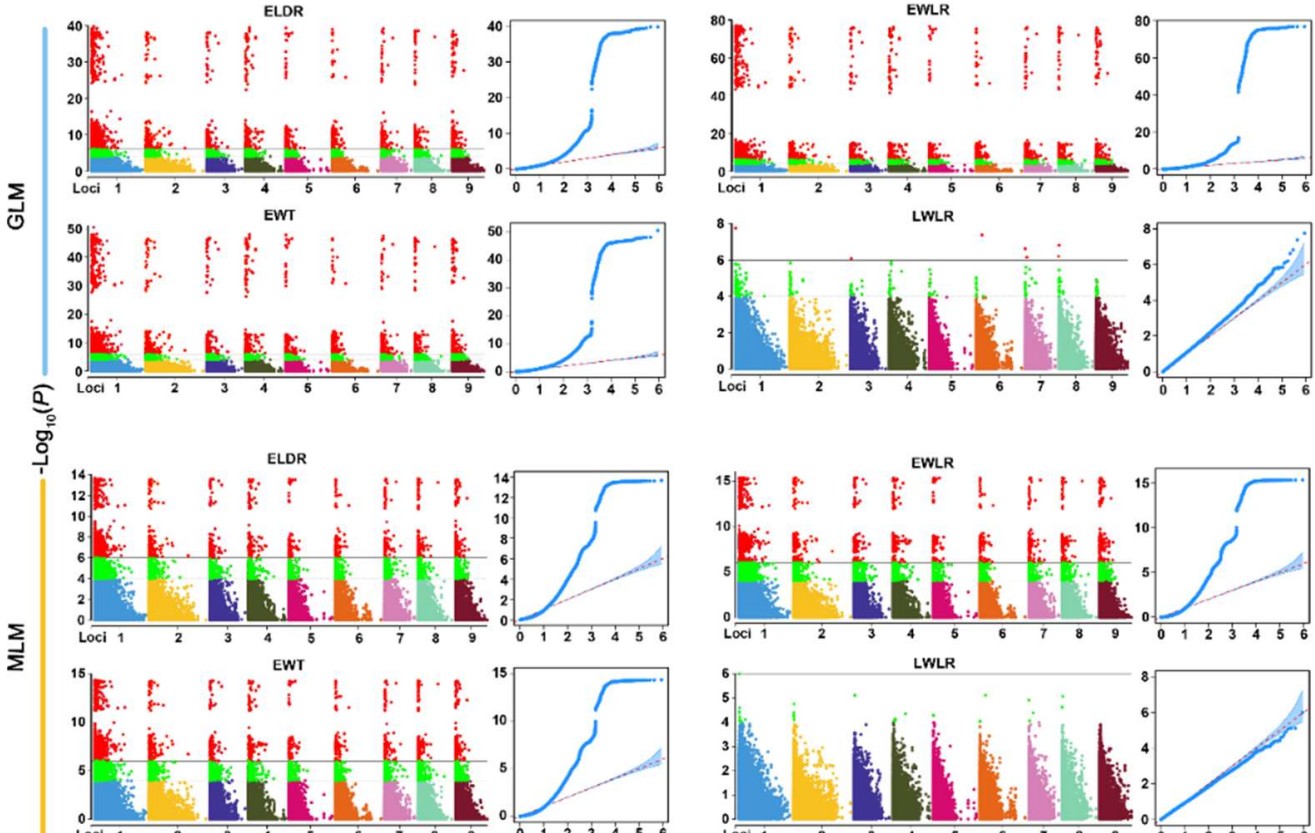

**Figure 5.** The Manhattan plots and QQ plots of four highly associated wood tracheid traits based on the GLM and MLM. The red points indicate the significantly associated SNP loci ($-\log_{10}(p)$ values > 6) and the green points indicate the moderately associated SNP loci ($6 > -\log_{10}(p)$ values > 4). Loci 1 indicates 323,277 loci (MA_1~MA_199993); Loci 2 indicates 78,521 loci (MA_2~MA_29998); Loci 3 indicates 65,557 loci (MA_3~MA_399967); Loci 4 indicates 75,052 loci (MA_4~MA_49999); Loci 5 indicates 52,270 loci (MA_5000~MA_5999942); Loci 6 indicates 55,725 loci (MA_60~MA_69999); Loci 7 indicates 61,650 loci (MA_7000058~MA_79999); Loci 8 indicates 77,603 loci (MA_8~MA_899996); and Loci 9 indicates 98,009 loci (MA_90~MA_9999978) in Norway spruce genome. The QQ-plots inset—right with observed $\log_{10}(p)$ values on the *y*-axis and expected $\log_{10}(p)$ values on the *x*-axis.

Two earlywood tracheid traits, EWLR and EWT, were associated with two candidate genes (MA_10303051g0010 and MA_10883g0010). The gene MA_10883g0010 was homologous with EXORDIUM-like 2 (EXL2), which may play a role in a brassinosteroid-dependent regulation of growth and development, and the extracellular EXORDIUM protein mediates cell expansion in Arabidopsis leaves [43,44]. Therefore, the homologous gene of MA_10883g0010 in *P. crassifolia* may be involved in earlywood tracheid cell expansion. Additionally, a number of genes related to metabolism and transportation were associated highly with EWLR, including carboxylase, reductases, phosphatase, and transporters. This suggests that various physiological and biochemical reactions and material transportation in tracheid probably affect the earlywood tracheid wall–lumen ratio.

**Table 3.** The candidate genes of significantly associated regions for three earlywood tracheid traits on the MLM.

| Trait | Gene ID | Loci | Gene-start | Gene-end | SNP location | Allele | Annotation |
|---|---|---|---|---|---|---|---|
| ELDR, EWLR, EWT | MA_10430313g0010 | MA_10428216 | 15911 | 16486 | 8522 | T/A | Serine/threonine-protein kinase |
| | MA_10429843g0020 | MA_10429843 | 27084 | 33339 | 14836 | G/T | Aminodeoxychorismate synthase |
| | MA_10434936g0010 | MA_10434936 | 3785 | 12707 | 16639 | T/C | Poly (ADP-ribose) glycohydrolase 1 |
| | MA_11137g0010 | MA_11137 | 25232 | 25570 | 17526 | G/A | Histone H1 |
| | MA_119933g0010 | MA_119933 | 7320 | 8147 | 15072 | C/A | Wall-associated receptor kinase |
| | MA_17692g0010 | MA_17692 | 82061 | 82593 | 87850 | G/A | Serine carboxypeptidase |
| | MA_19953g0020 | MA_19953 | 81799 | 91867 | 38054 | C/T | Pathogenesis-related protein 5 |
| EWLR, EWT | MA_10303051g0010 | MA_10303051 | 23536 | 23793 | 2008 | G/A | Multicopper oxidase |
| | MA_10883g0010 | MA_10883 | 53514 | 53744 | 17064 | G/A | EXORDIUM-like 2 |
| EWLR | MA_101170g0020 | MA_101170 | 24857 | 25888 | 32445 | C/T | Carboxylesterase 17 |
| | MA_10427040g0010 | MA_10427040 | 22428 | 23383 | 2504 | A/G | Heat shock 22 kDa protein |
| | MA_10427391g0010 | MA_10427391 | 21043 | 42330 | 58343 | G/A | MAG2 |
| | MA_10431643g0010 | MA_10431643 | 1 | 11637 | 12312 | A/G | Proteasome subunit alpha type-5 |
| | MA_10432762g0010 | MA_10432762 | 17888 | 18253 | 1144 | G/A | Telomerase reverse transcriptase |
| | MA_10433801g0010 | MA_10433801 | 551 | 15049 | 64877 | C/T | E3 ubiquitin-protein ligase |
| | MA_10437109g0030 | MA_10437109 | 94725 | 96103 | 91706 | G/C | ABC transporter D family member 1 |
| | MA_10437270g0010 | MA_10437270 | 42477 | 60307 | 1265 | G/T | IPG1 |
| | MA_114300g0010 | MA_114300 | 1314 | 3462 | 4171 | A/G | STRUBBELIG-RECEPTOR FAMILY 8 |
| | MA_174203g0010 | MA_174203 | 12346 | 13811 | 5106 | A/G | Sodium/metabolite cotransporter BASS1 |
| | MA_17795g0010 | MA_17795 | 42316 | 42738 | 57812 | T/A | Histone H2B.1 |
| | MA_182831g0010 | MA_182831 | 20010 | 21544 | 31133 | C/T | Hyoscyamine 6-dioxygenase |
| | MA_20154g0020 | MA_20154 | 67781 | 78419 | 57944 | C/T | Phosphoenolpyruvate carboxylase |
| | MA_31g0010 | MA_31 | 21393 | 40460 | 18526 | A/G | Ubiquitin carboxyl-terminal hydrolase 12 |
| | MA_34680g0010 | MA_34680 | 1 | 23548 | 847 | G/A | Peroxiredoxin-2F |
| | MA_39532g0010 | MA_39532 | 4950 | 6096 | 29400 | A/G | Copper transport protein CCH |
| | MA_417207g0010 | MA_417207 | 3307 | 5128 | 5643 | G/T | Purple acid phosphatase 3 |
| | MA_482451g0010 | MA_482451 | 1 | 1106 | 5136 | C/T | Ribulose-1,5 bisphosphate carboxylase |
| | MA_5201g0010 | MA_5201 | 4440 | 9434 | 12271 | G/A | Proline transporter 2 |
| | MA_588g0010 | MA_588 | 18671 | 26200 | 22351 | T/A | Abscisic acid hydroxylase 1 |
| | MA_74441g0010 | MA_74441 | 1775 | 11928 | 1579 | A/T | Aldo-keto reductase family 4 member C9 |
| | MA_80965g0010 | MA_80965 | 17882 | 22398 | 39315 | C/G | Flavin-containing monooxygenase 1 |
| | MA_951514g0010 | MA_951514 | 262 | 1867 | 3347 | A/G | 2-alkenal reductase |
| | MA_958g0010 | MA_958 | 40918 | 71141 | 41171 | G/T | MODIFIER OF SNC1 1 |
| | MA_99302g0010 | MA_99302 | 18687 | 19646 | 28542 | C/T | Thioredoxin-like protein CDSP32 |

## 4. Discussion

In typical GWAS, phenotype and genotype data are collected for a large sample of assembled individuals [45]. However, a representative sample size combined with high-throughput sequencing and appropriate algorithms is sufficient to generate a relatively rich set of SNP and association loci in forest trees [46,47]. While it is possible that a detected genetic marker resides within a causative gene for the phenotype of interest, this is often not the case. Instead, GWAS rely on linkage disequilibrium (LD) between markers under testing and functional polymorphisms of causative genes [48,49]. The large number of SNPs in GWAS of conifers species is a reflection of their large and complex genomes and dramatically increased marker density would enable said markers to better track LD with causal variants in these large, genetically diverse genomes [50]. A large number of SNP loci can be identified in relatively small number of samples in conifers. For example, in lodgepole pine, more than 95,000 SNPs were obtained in 98 serotinous and nonserotinous samples from three populations [51]. De la Torre et al. [52] identified 799 significant associations of cold-related traits by GWAS in 217 samples in Douglas-fir. GWAS

in 194 maritime pines from different families provided the map position of 1671 SNPs corresponding to 1192 different loci [39]. In our study, the studied populations were mostly composed of individuals representing a species with narrow distribution range [53], we intentionally selected 106 individuals representing different natural forest populations that could represent the core germplasm of Qilian Mountain as determined by the little genetic relationship among them. The results of GWAS showed that there are abundant phenotypic variation and associated SNP loci. Thus, these obtained results demonstrate that sample size is not the most important factor in GWAS and the genetic relationship among samples and LD effects should be the focus of sample selection. Therefore, samples representing the core germplasm resources can be used as the materials for association analysis of the quantitative traits under the limited materials (i.e., small sample size).

Genomes of conifers are huge and complex and only few species have their genomes sequenced including Norway spruce [41], white spruce [54], and loblolly pine [55]. Traditional methods in developing molecular markers are usually cumbersome, time-consuming, and in most cases are unable to meet the experiment requirements. SLAF-seq-based GWAS is a fast and cost-effective approach to develop a large number of SNP markers in the absence of reference genomes. It is an effective method to determine molecular markers that influence essential traits in the absence of whole-genome markers [56,57]. As a consequence, SLAF-seq-based GWAS has been used in several conifers, for example, Bai et al. [58] used the SLAF-seq technique to analyze *Pinus massoniana* Lamb. germplasm resources in Guangdong Province, and identified 471,660 SNP markers in 599,164 SLAF polymorphic markers. Yang et al. [59] also used this approach and detected 524,662 high-quality SLAFs and identified 249,619 SNPs in hybrid clones of Taxodium species. In this study, a total of 4,058,883 SLAF-tags were detected and 12,275,765 SNP markers were developed and employed in GWAS to identify SNP loci associated with important traits and determined their candidate genes.

GLM and MLM are the most used algorithmic models in GWAS. The advantage of GLM is that it is more comprehensive and can identify more SNPs associations with the traits; however, its accuracy is lower than that of MLM [60–63]. In our study, the associated analysis results showed that the significantly associated SNPs in MLM ($p < 1.11 \times 10^{-7}$) were actually less than that of the GLM. Meanwhile, in the analysis of associated candidate gene, latewood tracheid trait LWLR was associated with few candidate genes in GLM model, but not in MLM model (Figure 5). This is because MLM model is more stringent than GLM model, and MLM model considers the influence of population structure (Q value) and genetic relationship (K value) and removes possible false associations [64]. Therefore, MLM can improve the accuracy of the analysis but can also miss some important SNP loci due to the strict screening conditions. Thus, multiple algorithmic models should be used to conduct GWAS data analysis to overcome this limitation [65,66].

Additionally, a total of 14 wood tracheid traits of *P. crassifolia* were used in GWAS. Due to the large amount of genome data and high levels of genome wide heterozygosity, the Norway spruce genome we used as reference genome did not assign identified association to the chromosome level. As the majority of the SNP loci were associated with earlywood tracheid traits, this is an indication of extensive variations in earlywood formation and maybe the presence of a more complex regulatory network. In temperate zones, earlywood forms in the spring, and the activity of cell cambium and physiological metabolism are vigorous due to suitable temperature and sufficient water [67,68]. As a result, the development of earlywood tracheid in *P. crassifolia* is expected to harbor more variation and thus SNP loci were easily associated with earlywood tracheid traits. Hence, QQ plots of the four highly associated earlywood tracheid traits (ELDR, EWLR, EWT, and LWLR) were generated to validate the accuracy of the population correction. The results of the QQ plots showed that, overall, the observed values did not match the expected values except for a few outliers at the beginnings. In other words, the highly associated SNP loci do not show normal distribution in the three earlywood tracheid traits. If a SNP locus deviate from

expectation, it is considered that the deviation of the SNP observed value is caused by the genetic effect caused by the SNP mutation (i.e., true association) [69].

Next, candidate genes were screened in the 100 kb (50 kb upstream, 50 kb downstream) zone of each of the observed highly significant SNP loci. The MLM generated 67 potential candidate genes associated with three earlywood tracheid traits, ELDR, EWLR, and EWT (Figure 5). The multiple annotation databases results showed 34 candidate genes while the function of additional 33 candidate genes were unknown. The candidate gene MA_119933g0010 was associated with three earlywood tracheid traits (ELDR, EWLR, and EWT), and it is homologous with wall-associated receptor kinase. WAKs plays an important role in cell development, and our results indicated that the candidate SNP loci of gene MA_119933g0010 has a similar function in earlywood tracheid. Likewise, the gene MA_373300g0010 identified by the GWAS of the tracheid traits in Norway spruce is likely similar to WAKs [7]. In Arabidopsis, WAKs proteins are involved in cell wall expansion of leaf and regulation of leaf senescence [70–72]. Therefore, WAKs are probably key proteins and regulators in tracheid development of the spruce. The candidate gene MA_10883g0010 was associated with EWLR and EWT, and is homologous with EXORDIUM-like 2 that is active in mediating regulation of growth and development [43,44]. EXOORDIUM and WAKs proteins in *P. crassifolia* may have a significant effect in the development of earlywood tracheid. In addition, several candidate genes related to various enzymes and transporters were highly associated with EWLR, including carboxylesterase, phosphoenolpyruvate carboxylase, 2-alkenal reductase, ABC transporter D family member, sodium/metabolite cotransporter, and Proline transporter. The association of these enzymes and transporters also showed that metabolism and material transport are active in earlywood tracheid, and that the growth and development regulation of earlywood is important for *P. crassifolia*. We expect that these associated SNP loci with 67 candidate genes will provide essential genetic basis for earlywood tracheid traits improvement in *P. crassifolia*.

## 5. Conclusions

This work presents the first genome wide dissection of wood tracheid traits in *P. crassifolia*. A total of 67 highly significant SNP loci were associated with three earlywood tracheid traits, ELDR, EWLR, and EWT. These SNP loci have identified a set of candidate genes that could be exploited to improve earlywood tracheid traits for regulating development of earlywood tracheid for ecological adaptability of *P. crassifolia* under arid and semi-arid conditions. Previous research on the environmental effects on the development of tracheid in conifers mainly focused on morphology, and our study provided a major opportunity for understanding the underpinning of wood tracheid traits through using 12,275,765 SNPs for GWAS and extending the work to functional mapping approach. It is also worth mentioning that no associations were detected in latewood tracheid traits. The magnitude of variation earlywood tracheid traits was also higher than in latewood tracheid traits. These results showed that SNP mutations are more likely to affect growth and development of earlywood tracheid. Therefore, studies of wood development or drought adaptability of *P. crassifolia*, are more likely to benefit if focus is directed to the regulation and improvement of earlywood tracheid traits. Multiple SNP loci associated with the candidate genes related to cell and cell wall development are expected to provide a genetic basis for exploring and verifying the key mechanisms regulating earlywood tracheid development in *P. crassifolia*.

**Supplementary Materials:** The following supporting information can be downloaded at: https://www.mdpi.com/article/10.3390/f13020332/s1, Table S1: The information of 106 clones in *P. crassifolia*; Table S2: The sequencing result of *P. crassifolia*; Table S3: The SNP statistics of *P. crassifolia*; Table S4: The SLAF-tag statistics of *P. crassifolia*; Table S5: The information of SNP loci in *P. crassifolia*; Table S6: The locus information of reference genome; Table S7: The SRA metadata of SLAF-seq in *P. crassifolia*.

**Author Contributions:** Conceptualization, W.L., Y.A.E.-K. and C.Z.; methodology, C.Z., Y.G. and Y.C.; software, C.Z.; formal analysis, C.Z. and Y.G.; investigation, Y.C.; resources, H.Z.; writing—original draft preparation, C.Z.; writing—review and editing, W.L. and Y.A.E.-K.; visualization, C.Z.; and supervision, W.L. All authors have read and agreed to the published version of the manuscript.

**Funding:** The project was funded by the National Natural Science Foundation of China (31770713, 31860221).

**Institutional Review Board Statement:** Not applicable.

**Informed Consent Statement:** Not applicable.

**Data Availability Statement:** The raw sequencing data and the SLAF-sequencing data (SRA accession: SUB10523923), are available in the NCBI Sequence Read Archive (SRA) database under BioProject accession number PRJNA771805, the detailed information of SRA metadata is shown in Table S7. Other datasets supporting the conclusions of this article are included within the article and its additional files.

**Acknowledgments:** We are grateful for the generous grant from the National Engineering Laboratory of Tree breeding of Beijing Forestry University and Gansu Province Academy of Qilian Water Resource Conservation Forests Research Institute that made this work possible.

**Conflicts of Interest:** The authors declare no conflict of interest.

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
