# Peer review of "Genome Wide Association Study Identifies Candidate Genes Related to the Earlywood Tracheid Properties in Picea crassifolia Kom."

_forests, doi:10.3390/f13020332_

Round 1

Reviewer 1 Report

I found the manuscript satisfactory in content. However, I feel that the ms will require extensive English editing for the information conveyed by the authors to be clear. Below, find my comments.

Line 40: Remove” While”

Lines 43-46: Revise as follows:

“Studies conducted on P. crassifolia sampled from different altitudes of the Tibetan plateau, north-western China, found the development of tracheid radial diameter is closely related to temperature and precipitation, and trees could change their internal characteristics to adapt to changing climate”

Lines 53-54: Revise as follows:

however, no attempts have been made to understand the genetic factors underlying wood tracheid development variation.

Lines 64-71: It is not clear what is meant in this paragraph. Please revise

Although an earlier study identified quantitative trait loci (QTLs) for wood density variation in loblolly pine using linkage analyses from segregating family pedigrees, marker-aided selection (MAS) based on these QTL linkage analyses were less partially translated or unusually implemented in practical tree breeding due to QTLs are often family specific, generally explaining a small amount of phenotypic variation, and the need for very large QTL mapping families to recover desirable combinations of QTL leles for more than five or six loci [12-13]. And inconsistent associations among different families and the low transferability of markers [14

Lines 102-104: Revise as follows

”Clones were planted at 5 × 5 m within and between rows on the same soil type and managed similarly in a complete randomized block design with 18 replications”

Lines 128-129: Revise as follows

Clusters of libraries were were loaded into an Illumina HiSeq for paired-end sequencing

Line 132

Afterwards the SNP heterozygosity were calculated

Lines 139-140

Additionally, population structure analysis using 12,275,765 SNPs to infer the genetic background of clonal cluster membership under a given number of populations (K) was carried out.

Line 219. Confirm that this was the intended meaning and revise.

Kinship analyses also suggested that the population structure was simple (Figures 3D).

Lines 242:

The GLM and MLM models identified more significant SNPs for the former than the latter with some SNPs that overlapped between the two models.

Line 255

, yielding a total 255 of candidate genes were obtained – remove “obtained”

Line 301

Additionally, a series of genes. Replace “series” with “number”

Line 304:

in tracheid will affect the earlywood tracheid wall. Replace “will” with “probably”

Lines 308-309. Revise, it is not clear what the intended meaning was.

The relatively observed number of associations in the present study is comparable to other association studies of complex growth traits in forest trees and the number of samples

Lines 332. Revise as follows:

Genomes of cconifers are huge and complex and only a few species have their genomes sequenced including Norway spruce [41], white spruce [54], and loblolly pine [55].

Lines 347: Revise as follows:

GLM and MLM are the most used algorithmic models in GWAS. The advantage of GLM is that it is more comprehensive and can identify more SNPs associations with the traits, however, its accuracy is lower than that of MLM [60-63].

Line 353:

. This is because MLM model is stricter than GLM.. Replace “stricter” with “more stringent”

Lines 359-361. Confirm accuracy of intended meaning and revise as follows:

A total of 14 wood tracheid traits of P. crassifolia were used to used in the GWAS. Due to the large amount of genome data and high levels of genome-wide heterozygosity, the Norway spruce genome we used as reference genome did not assign identified association to the chromosome level.

Line 387-400. Confirm accuracy of intended meaning and revise as follows:

In Arabidopsis, WAKs proteins are involved in cell wall expansion of leaf, cell enlargement and regulation of leaf Senescence. Therefore, WAKs are probably key protein and regulators in tracheid development of the spruce. The candidate gene MA_10883g0010 was as associated with EWLR and EWT, and is homologous 390 with EXORDIUM-like 2 that is active in mediating regulation of growth and development 391 [43,44]. EXOORDIUM and WAKs proteins in P. crassifolia may have a significant effect in the development of earlywood tracheid. In addition, several candidate genes related to various enzymes and transporters were highly associated with EWLR, including carboxy- lesterase, phosphoenolpyruvate carboxylase, 2-alkenal reductase, ABC transporter D family member, Sodium/metabolite cotransporter, and Proline transporter. The association of these enzymes and transporters also showed that metabolism and material transport are active in earlywood tracheid, and that the growth and development regulation of earlywood is important for P. crassifolia. We expect that these association SNPs in the 67 candidate genes will provide essential genetic basis for earlywood tracheid traits improvement in P. crassifolia.

Reviewer 2 Report

The article Genome-Wide Association Study Identifies Candidate Genes Related to the earlywood tracheid properties in Picea crassifolia by Zhou et al. presents an evaluation of Genome-Wide Association for earlywood tracheid properties in Picea crassifolia accessions from china using the SLAF based SNP markers. It provides new information on candidate genes that have been associated with these traits. The manuscript is written well, and analyses have been carried out neatly.

Minor corrections required

Page 2 Line 53: “however, no attempts were made to understand the genetic factors underlying wood tracheid development variation”

Page 2 Line 61: Remove the comma after “wall-lumen diameter ratio,”

Reviewer 3 Report

The manuscript of Zhou et al describes the identification of 67 candidate genes related to earlywood tracheid traits in Picea through a SLAF-seq-based GWAS. 

The experimental set up is appropriate and the results are sound clearly presented. There are quite some small errors in the text that have to be corrected. 

Abstract:

Line 19-22. Make 2 phrases (at specifically)

Introduction:

Line 40 rephrase

Line 44: found that

Line 57: relevant instead of indispensable

Line 64-71: rephrase it is not clear

Line 75-76: what do you mean?

Material and methods

You start with Table 3. Check all table numbers and the text accordingly

Line 111: additionally

Line 132: was calculated

Line 134: remove one were

Line 139-141: rephrase

Results:

Please check all Table S: Table S1 is named S2, table S6 is S1 etc...

Table S6 and S7 are missing in the text

Line 182: more affected by the environment than??

Figure 1: enlarge

Table 1 is Table 2...

Line 247: contributed to more than 10% phenotypic

Line 249 remove the ,

Line 256 remove were obtained
